# Changes in Food Consumption, BMI, and Body Composition in Youth in the US during the COVID-19 Pandemic

**DOI:** 10.3390/ijerph20186796

**Published:** 2023-09-21

**Authors:** Nasreen Moursi, Marian Tanofsky-Kraff, Megan Parker, Lucy Loch, Bess Bloomer, Jennifer Te-Vazquez, Ejike Nwosu, Julia Lazareva, Shanna B. Yang, Sara Turner, Sheila Brady, Jack Yanovski

**Affiliations:** 1Section on Growth and Obesity, Division of Intramural Research, Eunice Kennedy Shriver National Institute of Child Health and Human Development (NICHD), National Institutes of Health (NIH), 10 Center Drive, Room 1-3330, Bethesda, MD 20892, USA; nasreen.moursi@usuhs.edu (N.M.); megan.parker.ctr@usuhs.edu (M.P.); lucyloc@umich.edu (L.L.); bess.bloomer@nih.gov (B.B.); jennifer.te-vazquez@nih.gov (J.T.-V.); ejike.nwosu@nih.gov (E.N.); julia.lazareva@nih.gov (J.L.); bradys@mail.nih.gov (S.B.); yanovskj@mail.nih.gov (J.Y.); 2Department of Medical and Clinical Psychology, Uniformed Services University of the Health Sciences (USUHS), Bethesda, MD 20814, USA; 3Nutrition Department, NIH Clinical Center, 10 Center Drive, Bethesda, MD 20892, USA; shanna.bernstein@nih.gov (S.B.Y.); sara.turner@nih.gov (S.T.)

**Keywords:** overweight, obesity, youth, COVID-19, pandemic, food consumption, BMI, adiposity

## Abstract

Rates of childhood overweight/obesity have risen for decades; however, data show the prevalence increased at a faster rate during the COVID-19 pandemic. Pandemic-associated increases in youth’s body mass index (BMI; kg/m^2^) have been attributed to decreases in reported physical activity; few studies have examined changes in food intake. We therefore examined changes in total energy, nutrient consumption, BMI, BMIz, and adiposity longitudinally over 3 years, comparing healthy youth aged 8–17 years assessed twice prior to the pandemic, to youth seen once before and once during the pandemic. The total energy intake and percent macronutrient consumption were assessed using a standardized, laboratory-based, buffet-style meal. Height and weight were measured and adiposity was collected via dual energy X-ray absorptiometry. Generalized linear model univariate analyses investigated differences between groups. One-hundred-fifteen youth (15.6 + 2.8 years 47.8% female; 54.8% White) from the Washington D.C., Maryland, and Virginia greater metropolitan area participated. In this secondary analysis, neither changes in total energy intake (*p* = 0.52) nor changes in nutrient consumption were significantly different between the two groups (*ps* = 0.23–0.83). Likewise, changes in BMI, BMIz, and adiposity (*ps* = 0.95–0.25) did not differ by group. Further research should investigate food intake and body composition, comparing youth with and without overweight/obesity to better identify those at greatest risk of excess weight gain during the pandemic.

## 1. Introduction

In March 2020, the SARS-CoV-2 virus, or COVID-19, spread throughout the United States, triggering lockdowns, social isolation, and quarantines. Since the beginning of the pandemic, studies have examined how the virus itself and subsequent societal changes have affected public health [1,2,3,4]. Early in the pandemic, youth and adults with overweight or obesity were identified as being at high risk for severe COVID-19 outcomes [5,6,7,8,9,10,11,12]. This spurred an effort to elucidate the link between overweight/obesity and COVID-19. In this effort, researchers found that beyond overweight/obesity being a risk factor for adverse COVID-19 outcomes, the pandemic, and/or the safety measures used to combat it, may be risk factors for excessive weight gain [7,13,14,15,16,17,18,19,20].

The prevalence of childhood obesity has been increasing in the United States for many decades [21], but since the start of the COVID-19 pandemic, prevalence has increased at a significantly higher rate; from 15.1% in 2018 to 15.7% in 2019 versus 17.3% in 2020 [16,18,22,23,24,25,26,27]. A systematic review and meta-analysis found consistent, significant increases in body weight and body mass index (BMI, kg/m^2^) for school-aged children during the pandemic and reported that the lockdown had a greater impact on weight gain among children compared to adults [22]. These increases in weight may be attributable to pandemic-related behavioral changes, such as reduced physical activity. Several studies have reported that increases in obesity during the pandemic were associated with reduced physical activity or increased sedentary activity and screen time [13,14,19,26,28]. At the beginning of the pandemic, many youths were confined to their homes, participating in remote learning and virtual social gatherings which resulted in increased sedentary and screen time. Moreover, many youths lost structured and social opportunities for physical activity, such as recess, physical education classes, playdates, or afterschool sports [13,14,19,26,28].

Decreased energy expenditures during the pandemic have been well documented, but there are less data on changes in energy intake. Studies examining energy intake during the pandemic in adult samples have yielded mixed findings. In one study, participants self-reported a significant increase in “unhealthy” food consumption behaviors during the pandemic compared to before the pandemic, as measured by the increased endorsement of “unhealthy” food intake, “out of control” eating experiences, snack intake, and consuming more than three main meals per day [13]. In another study, researchers identified six healthy and unhealthy dietary patterns and associated them with positive and negative changes in physical activity [29]. For example, the 29.5% of participants who reported reducing their physical activity during the pandemic had higher mean factor scores for the rice-pasta-chicken dietary pattern (increasing intake of rice, pasta, chicken, eggs, and legumes), which was a relatively healthy dietary pattern compared to the other patterns identified [29]. In a survey of young adults, 31.2% reported their overall food intake increased and yet 16.8% reported their food intake decreasing since the pandemic started, with increased food consumption being more likely reported by people with higher BMIs [30]. In a third study, 40% of adults reported gaining weight during the pandemic and that those individuals were more likely to endorse increasing consumption of unhealthy foods such as ultra-processed food and unhealthy snacks [15].

Research related to food consumption during the pandemic in pediatric samples is sparse. One study examined youth under lockdown in the early pandemic and found that youth reported increasing their fruit, chips, red meat, and sugary drink intake, but there was no change in their intake of vegetables [28]. In 2021, another study on lifestyle changes in youth under lockdown found an increase in reported consumption of fruit, fruit juice, vegetables, dairy, pasta, sweets, snacks, and breakfast foods and a decrease in fast foods [14]. The authors noted that the 35% of youth who reported an increase in body weight were more likely to endorse consuming more breakfast foods, salty snacks, and overall snacks [14]. These studies demonstrate the wide range of healthy (e.g., increasing vegetables) and less healthy (e.g., increasing sweets) food consumption changes recorded in youth during the pandemic. 

Although the aforementioned studies provide initial insight into the associations between food consumption, weight, and the COVID-19 pandemic, there is a need for more research on energy intake in youth and body size, particularly using objective measures. The aim of this study was to objectively measure changes in food consumption, BMI, and body composition in healthy youth before and during the COVID-19 pandemic. Specifically, we investigated (a) the changes in total energy and percent nutrient energy consumption during a laboratory-based test meal and (b) the changes in BMI, BMIz, and adiposity over a 3-year period, comparing youth assessed twice prior to the pandemic (No Pandemic group) to youth assessed once before to the pandemic and once during the pandemic (Pandemic group). We hypothesized that (a) the Pandemic group would have greater increases in total energy consumption and (b) greater increases in BMI, BMIz, and adiposity than the No Pandemic group. No hypothesis was made for percent nutrient consumption given the mixed findings regarding the change in foods consumed during the pandemic in prior data.

## 2. Materials and Methods

### 2.1. Participants

The current study is a secondary analysis of an ongoing, longitudinal study (ClinicalTrials.gov ID: NCT02390765) investigating biopsychosocial factors associated with disinhibited eating behaviors and excessive weight gain among youth. This longitudinal study began in 2015. Some food intake findings from this sample have been reported previously; however, none have investigated the changes in food consumption, BMI, and body composition in relation to the COVID-19 pandemic. Youth were generally healthy boys and girls aged 8–17 years at baseline recruited from Washington D.C., Maryland, and Virginia via direct mailings to families within the greater metropolitan area of the National Institutes of Health (NIH), flyers posted at the NIH and other local facilities (i.e., libraries, supermarkets, etc.), and advertisements and flyers posted online and on social media. All recruitment efforts were in the English language and targeted to families with youth in the appropriate age range. Youth were excluded if they were underweight (BMI < 5th percentile for age and sex), had significant weight loss in recent months (>5% of body weight), had a history of recent concussion or brain injury, were currently taking medications or substances known to impact weight or eating behaviors, had a Full Scale Intelligence Quotient ≤ 70 on the Wechsler Abbreviated Scale of Intelligence (WASI-II) [31], or had a diagnosis of a serious medical condition or full threshold psychiatric disorder as defined by the Diagnostic and Statistical Manual—5th Edition [32], other than a binge-eating disorder. The presence of exclusionary psychiatric conditions was assessed at baseline by the Kiddie Schedule for Affective Disorders and Schizophrenia for School-Age Children [33]. 

Potential participants accompanied by a parent or guardian completed eligibility screening and baseline study visits at the National Institutes of Health Hatfield Clinical Research Center in Bethesda, MD. Youth and their parents/guardians provided informed assent and consent, respectively, before study procedures were administered. All study procedures were approved by the National Institutes of Health (NIH) institutional review board. Per protocol, approximately three years after completing their baseline visit, participants and their parents/guardians returned to the NIH to repeat all measurements.

Sample selection was based on data completion and the dates of assessment. First, youth with incomplete data on any measures for either the baseline or 3-year visit were excluded. Second, youth were grouped into No Pandemic and Pandemic groups by the dates of their baseline and 3-year visits (Figure 1). The No Pandemic group consisted of all participants who completed both visits prior to 6 March 2020. The Pandemic group consisted of participants who completed a baseline visit before 6 March 2020 and a 3-year visit on or after 27 August 2020. There were no data collected between 6 March and 26 August 2020, due to COVID-19 safety restrictions on human subject research at the National Institutes of Health (NIH) Institutional Review Board.

### 2.2. Measures

To assess total intake of energy (kcal) and percentage nutrient intake (relative to total energy intake), youth were provided a standardized test meal. To standardize participants’ hunger levels prior to the meal, youth were instructed to fast overnight prior to the visit. Participants were given a standardized breakfast shake (17% protein, 16% fat, 67% carbohydrates) at around 10 A.M. which provided 21% of their estimated daily energy needs based on their measured body weight, height, age, and average activity level. At approximately 12:30 P.M., participants were served a buffet-style test meal (>10,000 kcal) which was composed of 12.1% protein, 32.8% fat, and 55.1% carbohydrates. Participants were instructed to “Let yourself go and eat as much as you want” [34]. The experimenter left the room until the participant indicated they were finished. All foods were weighed on electronic balance scales to the nearest gram. Food intake (g) was determined by subtracting post-meal food weights from pre-meal food weights. Energy intake and nutrient content of the consumed meals were calculated from food intakes using nutrient composition values from the USDA National Nutrient Database for Standard Reference [34] and ProNutra software version 3.6.04. The nutrients calculated were protein, fat, carbohydrate, fiber, sugar, saturated fatty acids, and monounsaturated fatty acids. To ensure consistency in the method and presentation of the buffet, the nutrition staff who prepared and measured the food completed yearly food safety training and had experience with processing food for controlled feeding studies. The procedure for this standardized test meal has been previously reported and validated as a measure of disinhibited eating in youth [34].

Height (cm) and fasting weight (kg) were assessed at a physical exam and used to calculate BMI (kg/m^2^) and BMIz scores. Height was measured in triplicate on an electronic, calibrated stadiometer (Holtain Ltd., Crymych, Wales, UK) and fasting weight was measured using a scale calibrated to 0.1 kg. The average of the three height measurements and measured weight were used to calculate BMI. BMIz scores were computed according to the CDC standards for age and sex [35]. Adiposity (percent fat mass) and lean mass (kg) were measured using dual energy X-ray absorptiometry (iDXA system, GE Healthcare, Madison, WI, USA), which is a validated body composition measure in youth [36]. The iDXA scan is a non-invasive measurement that utilizes low-dose X-rays and requires the participant to lay flat on their back for 10–20 min as the scanning arm slowly moves across the body [37].

### 2.3. Data Analysis Plan

All analyses were conducted using IBM SPSS 28.0.1. Data were screened for normality and for extreme but plausible outliers (>3 SD from the mean). Difference scores were calculated as the baseline value subtracted from the 3-year visit value. Differences scores for total energy intake, percent nutrient consumed, BMI, and BMIz and adiposity were all reciprocal transformed to improve normality. Independent *t*-tests and chi-square tests were conducted to assess group differences in covariates including age at baseline (years), sex assigned at birth, race, ethnicity, height, baseline BMI, baseline BMIz, baseline lean mass (kg), baseline adiposity, and time between visits (months), as appropriate.

Eleven Generalized Linear Models (GLMs) were conducted to investigate group differences (No Pandemic v. Pandemic) in the dependent variables; change in total energy intake, percent nutrients consumed, BMI, BMIz, and adiposity. The seven nutrients assessed were protein, fat, carbohydrates, fiber, sugar, saturated fatty acids, and monounsaturated fatty acids. For each dependent variable, one fully adjusted model with all appropriate covariates was conducted. If the fully adjusted model was significant, another partially adjusted model with only significant covariates was run. Fully adjusted models are reported below unless otherwise specified. Since the analyses were a series of pairwise comparisons, Bonferroni Hochberg corrections were used to adjust for multiple tests. The fully adjusted models for changes in total energy intake and changes in all seven nutrient intakes included covariates for age, sex assigned at birth, height, race, ethnicity, lean mass, adiposity, and time between visits. The fully adjusted model for changes in BMI included covariates age, sex, race, ethnicity, time between visits, and baseline BMI. The fully adjusted model for changes in BMIz included covariates height, race, ethnicity, time between visits, and baseline BMIz. The fully adjusted adiposity model was adjusted for age, sex, height, race, ethnicity, time between visits, and baseline adiposity. 

## 3. Results

### 3.1. Participants

Out of the 129 youth who completed a baseline and 3-year visit, 115 youth (89.2%) were included in the analyses (Table 1). Of the 14 excluded participants, 13 were excluded for not completing the test meal and one participant was excluded for not completing the DXA scan; chi-square found no significant differences in sex, race, ethnicity (*ps* = 0.15–0.93) and *t*-tests found no significant differences in age, baseline BMI, and adiposity (*ps* = 0.18–0.84) between excluded and included participants. The sample had a mean age of 12.4 years at baseline (SD = 2.7), was 47.8% female, 54.8% White, and 93% Non-Hispanic/Latino. The No Pandemic group consisted of 72 youths and the Pandemic group consisted of 43 youths. There were no significant differences between groups at baseline about age, sex assigned at birth, race, ethnicity, lean mass, adiposity, BMI, BMIz, or total energy intake. Significant differences between groups were observed in the time between visits such that the Pandemic group (M = 39.5 months, SD = 3.9) had more time between their baseline and 3-year visits than the No Pandemic group (M = 37.9 months, SD = 2.5, t(63.08) = −2.48, *p* = 0.02). This difference was most likely due to the COVID-19 lockdown since there was a period between March and August of 2020 when the research team was unable to collect data for safety reasons.

### 3.2. Within-Group Changes between Baseline and 3-Year Visits

Difference scores were calculated for total energy intake, BMI, BMIz, and adiposity. Using paired-sample *t*-tests, the No Pandemic group had significant increases in energy intake and BMI (t(71) = −2.85, *p* = 0.005; t(71) = −6.96, *p* < 0.001), and the Pandemic group had significant increases in energy intake (t(42) = −3.35, *p* = 0.001), BMI (t(42) = −3.20, *p* = 0.003), and significant decreases in BMIz (t(42) = 2.37, *p* = 0.021) and adiposity (t(42) = 1.99, *p* = .050). See Appendix A. However, when the mean difference scores for each group were compared in a fully adjusted GLM, there were no significant group differences in the within-group changes in total energy intake, BMI, BMIz, or adiposity. See Appendix A). These results reflect the natural and expected trend for youth to gain weight and consume more calories as they get older and gain height, and as the adjusted models demonstrate, when controlling for this growth trend, there were no significant differences in the change in total energy intake, BMI, BMIz, or adiposity between the baseline and 3-year visits in youth examined before the pandemic versus youth examined during the pandemic.

### 3.3. Total Energy Intake and Percent Nutrients Consumed

Adjusting for all covariates, there was no significant difference in the change in total energy consumed between the No Pandemic and Pandemic groups, (F(1, 104) = 1.57, *p* = 0.13). See Table 2.

Likewise, the No Pandemic and Pandemic groups did not significantly differ in the change in the percentage of energy consumed from protein (F(1, 104) = 1.18, *p* = 0.32), fat (F(1, 104) = 1.43, *p* = 0.19), carbohydrates (F(1, 104) = 1.06, *p* = 0.40), fiber (F(1, 104) = 0.73, *p* = 0.68), sugar (F(1, 104) = 1.29, *p* = 0.25), saturated fatty acids (F(1, 104) = 1.24, *p* = 0.28), or monounsaturated fatty acids (F(1, 104) = 1.66, *p* = 0.12) out of total energy consumed. No fully adjusted models were significant. Nutrient consumption models are reported in Appendix A.

### 3.4. BMI and Body Composition

When comparing the change in the BMI, BMIz, and adiposity from baseline to 3-year visits, there were no significant differences between the No Pandemic and the Pandemic groups. The fully adjusted models for change in BMI and BMIz were significant (F(1, 94) = 3.65, *p* = 0.002; F(1, 96) = 2.36, *p* = 0.036), but the group was not a statistically significant predictor of change in BMI (*p* = 0.95; Table 3) or change in BMIz (*p* = 0.38; Table 4). There was no significant difference in change in adiposity between groups (*p* = 0.25; Table 5).

## 4. Discussion

In this longitudinal examination of between-subject differences in 3-year change in total energy intake, nutrients consumed, BMI, BMIz, and adiposity, youth’s outcomes did not significantly differ in terms of whether they were studied prior to or during the COVID-19 pandemic. Based on research showing that youth were rapidly gaining weight during the early pandemic [16,18,22,23,24,25,26,27] and some (but not all) data suggesting food intake in adults may have increased [13,15,29,30], we theorized that an increase in youth’s food intake would have contributed, at least partially, to the reported weight gain. However, our findings are not consistent with our hypotheses. 

The previous research on changes in food intake in adults during the pandemic has mostly been based on self-report surveys administered after the onset of the pandemic [13,15,29,30]. Thus, pre-pandemic food intake was based on participant recall compared to their current eating patterns. Further, studies focused on youths relied not only on retrospective recall, but also on parent-report of their youth’s eating patterns [14,28]. Thus, prior reports may have been influenced by subjective bias due to recall surveys. Our study objectively measured food intake at a buffet-style meal both before and during the pandemic in a controlled environment. This methodological difference may have led to our null findings that contradict previous research. Additionally, the current study was a secondary analysis based on a buffet-style meal that was originally designed to induce loss-of-control eating or the experience of being unable to stop eating [34]. Studies have shown an association between loss-of-control eating and increased intake of energy-dense foods such as carbohydrates, sugary desserts, and other highly palatable foods and decreased intake of proteins [34,38]. Thus, the meal paradigm may have encouraged all youth to overeat at all visits. This could have created a ceiling effect explaining why we were unable to observe differences in food intake between the No Pandemic and Pandemic groups.

Interestingly, we were also unable to replicate findings that youth gained more BMI, BMIz, and adiposity in the Pandemic versus the No Pandemic group. Our hypothesis that youth assessed during the pandemic would gain more weight and adiposity than youth examined before the pandemic was based on literature demonstrating that youth gaining excess weight shortly after the onset of the pandemic [16,18,22,23,24,25,26,27] was in part due to a reduction in their physical activity in the same time period [13,14,19,26,28]. However, more recent data suggest that there was a rapid increase in the rates of obesity at the beginning of the pandemic that slowed later in the pandemic [35]. Our sample of pandemic data ranged from August 2020 to October 2021, thereby missing the first several months of the pandemic due to COVID-19 safety restrictions. Our sample was healthy and contained few individuals with very low socioeconomic status and it may well be that those with underlying medical or psychological conditions and/or the lowest disposable income were most susceptible to gaining weight during the pandemic. Nevertheless, it is also possible that our objective measures of excess weight gain in youth, done in the research setting were more accurate than previous early pandemic research, which often was based on self-reported heights and weights (though some objective data also show increases in obesity prevalence [16,18,23]). Moreover, adiposity based on DXA is considered superior to BMI, which cannot differentiate between muscle/lean mass and fat mass [39]. Our study did not assess changes in physical activity levels and it is possible that individual differences in energy output or sedentary activity during the pandemic could have affected the BMI and adiposity outcomes.

Recent data on the impact of the pandemic suggest that youth and adults with pre-existing overweight/obesity prior to the pandemic were more likely to gain weight during the pandemic than people without pre-existing overweight/obesity [20,22,26,30]. Overweight/obesity is a risk factor for major depressive disorder and anxiety disorder in youth [40,41,42,43] and prior to the pandemic, stressful life events and increased negative affect such as anxiety and depression have been associated with disinhibited eating behaviors in youth [34,38,44,45,46,47,48,49,50]. It is possible that youth with overweight/obesity were more vulnerable to stress-related factors promoting increased food intake specifically triggered by pandemic-related factors [34,38,46,48,51,52,53]. Specifically, abrupt societal changes such as lockdowns, school closures, and social isolation might have led to a reliance on eating as a coping mechanism for emotional distress, thereby causing an increase in food consumption and excessive weight and adiposity gain [34,38,44,45,46,47,48,49,50]. Foods that characterize disinhibited eating (e.g., highly palatable sweets and desserts) may be a mechanism promoting weight gain [38,54,55]. This could be an important connection in the relationships between societal stressors, affect-triggered loss-of-control eating, and increased food consumption, BMI, and adiposity that warrants future research. 

Data on individual differences in the impact of the pandemic (e.g., access to food, job security, education disruption or COVID-19 infection) were not assessed. Yet, youth’s experience of societal upheaval, social isolation, and access to food during the pandemic may have contributed to poor weight management or weight gain. Social isolation has been associated with sedentary behaviors, reduced physical activity, and increased food consumption [56,57,58]. Additionally, previous studies have shown that social support is associated with improved weight management [56,57,58]. Food choice is another complex variable that may have been affected by pandemic-related factors, such as changes in food security and food access [59,60,61,62,63]. Therefore, considering these factors and their relation to eating behaviors and weight changes during the pandemic may help elucidate important contexts for weight gain during the pandemic. 

Study strengths include objective measurements of food consumption, BMI, and body composition variables by trained staff. The use of longitudinal data to compare the changes over time in the Pandemic and No Pandemic groups was another strength. This is a difference from previous literature relying on cross-sectional retrospective recall with only one time point per group and it allowed for examination of change in food consumption, BMI, and adiposity. Our choice of a between-subjects design was supported in that there were no baseline differences in BMI, BMIz, or total energy intake. However, youth are expected to have individual differences in growth over a 3-year period, and thus future research should consider examining retroactively within-subject differences before and during the pandemic.

The current study has several limitations. First, the original study was designed and implemented in 2015. Thus, no assessments concerning COVID-19 or pandemic outcomes were included, such as individual differences in severe outcomes and negative life events. For example, one participant may have had both parents lose a job and attended school virtually for a whole year, whereas another participant may have had a parent who worked from home before the pandemic and was homeschooled without interruption. Second, due to safety restrictions, the study team was unable to collect data between 26 March and 26 August 2020. The early months of the pandemic may have been the most isolating and stressful given government lockdowns, schools shutting down and switching to virtual learning, furloughs and layoffs, and major changes in food access. Therefore, the timing of the greatest impact on food consumption, BMI, and adiposity may have been missed. Third, the test meal paradigm is a unique tool for objectively measuring total and nutrient consumption at a single point in time. However, its standardized nature and laboratory administration may not generalize to real-world eating experiences. Fourth, the sample size was limited to 115 youth due to an inability to collect additional data from No Pandemic children. Finally, it is possible that we did not observe greater increases in weight and BMI among the Pandemic group compared to the No Pandemic group in the current study because only 24% of the cohort was overweight/obese before the COVID-19 pandemic, and thus our participants were perhaps less likely vulnerable to stressors that promote overeating. The sample size of youth with overweight or obesity was too small (n = 28) to carry out follow-up analyses to test this hypothesis. 

## 5. Conclusions

We found minimal differences in changes in food consumption, BMI, and adiposity over 3 years between youth who were assessed prior to and during the pandemic. This contradicts some previous self-report research on the impact of the pandemic on eating behavior and weight. However, we studied generally healthy youth across the weight spectrum, and some have theorized that youth with overweight/obesity were more affected by the pandemic in terms of worsening their food consumption, BMI, and adiposity. Importantly, the pandemic has not been a homogenous stressor. Individuals were not impacted equally, and the early months of the pandemic may have experienced different influences than the later months. When retrospectively assessing the effects of the pandemic on weight gain, future research should consider selecting samples of youth with overweight/obesity and/or those at risk of developing obesity due to familial predisposition. Understanding these long-term impacts could guide intervention methods towards targeting youth whose dietary patterns, body weight, and adiposity were most affected by the COVID-19 pandemic.

## Figures and Tables

**Figure 1 ijerph-20-06796-f001:**
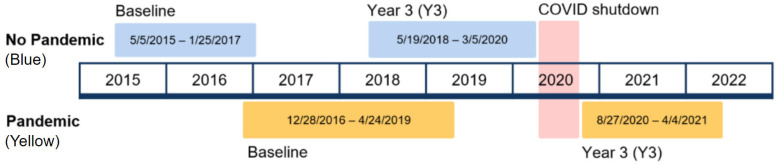
Timeline of study visits with No Pandemic and Pandemic groups. Note: No Pandemic group is shown in blue and the Pandemic group is shown in yellow. The red area represents the time when data collection was halted by COVID-19 safety restrictions.

**Table 1 ijerph-20-06796-t001:** Descriptive statistics with independent *t*-tests and chi-square analyses.

Variable	Total (*n* = 115)	No Pandemic Group (*n* = 72)	Pandemic Group (*n* = 43)	t	*p*
	M, SD	M, SD	M, SD		
Age (years)	12.4, 2.7	12.3, 2.8	12.6, 2.7	−0.61	0.54
BMI (kg/m^2^)	21.4, 5.4	20.8, 4.7	22.5, 6.2	−1.67	0.09
BMIz	0.56, 1.00	0.47, 0.94	0.73, 1.10	−1.35	0.18
Lean Mass (kg)	36.5, 11.9	35.4, 12.0	38.2, 11.6	−1.21	0.23
Adiposity (% fat mass)	0.55, 0.10	0.54, 0.10	0.57, 0.10	−1.37	0.09
Time between Visits (months)	38.5, 3.2	37.9, 2.5	39.5, 3.9	−2.48	0.02 *
Total Energy Intake (kCal)	986.8, 418.3	943.2, 425.1	1059.8, 400.8	−1.4	0.15
	N, %	n, %	n, %	χ^2^	*p*
Sex Assigned at Birth (female)	55, 47.8	35, 48.6	20, 46.5	0.05	0.83
Race:				4.91	0.30
White	63, 54.8	36, 50.0	27, 62.8		
Black or African-American	32, 27.8	25, 34.7	7, 16.3		
Asian	10, 8.7	5, 6.9	5, 11.6		
Multiple Races	8, 7.0	5, 6.9	3, 7.0		
Unknown	2, 1.7	1, 1.4	1, 2.3		
Ethnicity:				3.31	0.19
Non-Hispanic or Latino	107, 93.0	66, 91.7	41, 95.3		
Hispanic or Latino	7, 6.1	6, 8.3	1, 2.3		
Unknown	1, 0.9	0, 0	1, 2.3		

Note: BMI = body mass index; BMIz = standardized BMI score on a z-distribution; N = sample size; M = mean; SD = standard deviation; t = *t*-test statistic; *p* = significance value; χ^2^ = chi square statistic. * Significant assuming α = 0.05.

**Table 2 ijerph-20-06796-t002:** Change in total energy intake (kcal) with covariates.

	Fully Adjusted		Partially Adjusted	
	F	*p*	F	*p*
Model	1.57	0.13	1.96	0.15
Group	0.41	0.52	0.62	0.43
Age	2.20	0.14	--	--
Sex Assigned at Birth	2.02	0.16	--	--
Height	1.33	0.25	--	--
Race	0.01	0.94	--	--
Ethnicity	0.50	0.48	--	--
Time Between Visits	1.00	1.00	--	--
Lean Mass	0.92	0.34	--	--
Adiposity	6.18	0.02 *	2.89	0.09

Note: Group was operationalized as No Pandemic (0) and Pandemic (1). * Significance assuming α = 0.05.

**Table 3 ijerph-20-06796-t003:** Change in BMI with covariates.

	Fully Adjusted		Partially Adjusted	
	F	*p*	F	*p*
Model	3.65	0.002 *	6.47	<0.001 *
Group	<0.01	0.95	0.01	0.94
Age	14.26	<0.001 *	16.48	<0.001 *
Sex Assigned at Birth	<0.01	0.97	--	--
Race	0.39	0.54	--	--
Ethnicity	0.03	0.87	--	--
Time Between Visits	5.99	0.02 *	6.31	0.01 *
Baseline BMI	11.93	<0.001 *	13.06	<0.001 *

Note: Group was operationalized as No Pandemic (0) and Pandemic (1). * Significance assuming α = 0.05.

**Table 4 ijerph-20-06796-t004:** Change in BMIz with covariates.

	Fully Adjusted		Partially Adjusted	
	F	*p*	F	*p*
Model	2.36	0.04 *	4.16	0.02 *
Group	0.80	0.38	1.43	0.24
Race	1.59	0.21	--	--
Ethnicity	0.66	0.42	--	--
Height	2.12	0.15	--	--
Time Between Visits	0.33	0.57	--	--
Baseline BMIz	4.11	0.045 *	5.96	0.02 *

Note: Group was operationalized as No Pandemic (0) and Pandemic (1). * Significance assuming α = 0.05.

**Table 5 ijerph-20-06796-t005:** Change in adiposity (% fat mass) with covariates.

	Fully Adjusted		Partially Adjusted	
	F	*p*	F	*p*
Model	1.63	0.13	4.20	0.01 *
Group	1.37	0.25	1.42	0.24
Age	0.02	0.90	--	--
Sex assigned at Birth	6.29	0.01 *	7.21	0.01 *
Height	0.03	0.86	--	--
Race	0.54	0.47	--	--
Ethnicity	0.05	0.82	--	--
Time Between Visits	6.90	0.01 *	7.35	0.01 *
Baseline Adiposity	1.63	0.13	4.20	0.01 *

Note: Group was operationalized as No Pandemic (0) and Pandemic (1). * Significance assuming α = 0.05.

## Data Availability

The data presented in this study are available upon request.

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
