# Peer review of "Changes in Food Consumption, BMI, and Body Composition in Youth in the US during the COVID-19 Pandemic"

_ijerph, 2023, doi:10.3390/ijerph20186796_

Round 1

Reviewer 1 Report

Dear Authors,

the manuscript is clear, interesting and well written.

I suggest adding the geographical reference of the study in the title and abstract.

It is necessary to add information on the sampling of participants: how and where they were selected, information on the characteristics of waste and participants who were not included in the analyses, etc.

Line 145: the sum of protein, fat and carbohydrates does not make 100.

Author Response

Reviewer 1 Comments:

The manuscript is clear, interesting, and well written.

We thank the reviewer for this positive feedback.

I suggest adding the geographical reference of the study in the title and abstract.

We have added the geographical location to the abstract of the manuscript as (page 1) “…the Washington D.C, Maryland, and Virginia greater metropolitan area.” On page 1, we changed the title of the manuscript to “Changes in Food Consumption, BMI, and Body Composition in US Youth during the COVID-19 Pandemic”.  

It is necessary to add information on the sampling of participants; how and where they were selected, information on the characteristics of waste and participants who were not included in the analyses, etc.

Thank you for this suggestion. On page 3, the following information on recruitment efforts was added to the first paragraph of the Methods section: “Youth were generally health boys and girls ages 8-17 years at baseline recruited from the Washington D.C., Maryland, and Virginia via direct mailings to families within the greater metropolitan area of the National Institutes of Health (NIH), flyers posted at the NIH and other local facilities (i.e., libraries, supermarkets, etc.), and advertisements and flyers posted online and on social media. All recruitment efforts were in the English language and targeted to families with youth in the appropriate age range.”

In the first sentence of the results section (page 5), “Out of the 129 youth who completed a baseline and 3-year visit, 115 youth (89.2%) had complete data and were included in analyses (Table 1).” Additionally, we have added information on the excluded participants (page 5): “Of the 14 excluded participants, 13 were excluded for not completing the test meal and one participant was excluded for not completing the DXA scan; chi-square found no significant differences in sex, race, ethnicity (ps = .15-.93) and t-tests found no significant differences in age, baseline BMI, and adiposity (ps = .18-.84) between excluded and included participants.”

We have included the following information in the third paragraph of the methods section to clarify the sampling procedure: “Sample selection was based on data completion and the dates of assessment. First, youth with incomplete data on any measures for either the baseline or 3-year visit were excluded. Second, youth were grouped into No Pandemic and Pandemic groups by the dates of their baseline and 3-year visits (Figure 1).”

At the time of the data analysis, all youth with complete data fit the requirements of the assessment dates, thus, all 115 youth were categorized into the No Pandemic and Pandemic groups.

Line 145: the sum of protein, fat and carbohydrates does not make 100.

We thank the reviewer for noticing this discrepancy. For the test meal, the percentages of the macronutrients differ very slightly by each participant because the buffet has items such as “18 chicken nuggets” and “3 oranges”. The weights of these items will have some small variability meal to meal, which impacts the exact percentages of nutrients. Currently, using default weights of all the food items (versus weighing out the 3 oranges every time, etc.) the percentages add up to 12.1% protein, 32.8% fat and 55.1% carbohydrate.

We have changed the percentages in the manuscript to include the correct decimal places so the numbers do add to 100% (page 4): “At approximately 12:30 P.M., participants were served a buffet-style test meal (>10,000 kcal) which was composed of 12.1% protein, 32.83% fat, and 55.14% carbohydrates.”

Reviewer 2 Report

The authors must be commended for carrying out a study regarding food consumption and body composition of youth during COVID-19 pandemic. This topic is still relevant, and it is very important. The research methodology used in the study is appropriate, and the manuscript is written with good clarity. However, some issues need to be taken into consideration.

Introduction

First paragraph, second and last sentence: Please add a reference.

Line 89: Replace ''To this end'' with ''The aim of this study..''.

Methods

Line 139: Please add a study regarding the validity of the meal test.

Line 159: Please add a manufacturer of the stadiometer.

Please elaborate on the measurement of iDXA.

Discussion

Second paragraph, the first sentence: Please add a reference.

A very important part in the prevention of obesity is physical activity. I think one of the reasons for the non-existence of differences between groups is the level of physical activity during the pandemic. Some participants may have practiced regular physical activity at home during the lockdown, which may influenced their BMI, adiposity etc. You should mention that in the discussion. Also, I think this is one of the study’s limitations, you don’t have data on the physical activity level of the participants during the before and during the pandemic.

Author Response

Reviewer 2 Comments:

The authors must be commended for carrying out a study regarding food consumption and body composition of youth during COVID-19 pandemic. This topic is still relevant, and it is very important. The research methodology used in the study is appropriate, and the manuscript is written with good clarity. However, some issues need to be taken into consideration.

Thank you for your positive appraisal of our manuscript. We appreciate your feedback!

Introduction

First paragraph, second and last sentence: Please add a reference.

The appropriate citations were added to the sentences in question (page 1); “Since the beginning of the pandemic, studies have examined how the virus itself and subsequent societal changes have affected public health [1-4]… In this effort, researchers found that beyond overweight/obesity being a risk factor for adverse COVID-19 outcomes, the pandemic, and/or the safety measures used to combat it, may be risk factors for excessive weight gain [7,13-20].”

Line 89: Replace ''To this end'' with ''The aim of this study...''.

We have replaced “To this end…” such that the sentence now reads (page 2) “The aim of this study was to objectively measured changes in food consumption, BMI, and body composition in healthy youth before and during the COVID-19 pandemic.”

Methods

Line 139: Please add a study regarding the validity of the meal test.

We have added the following information at the end of the same paragraph on page 4: “The procedure for this standardized test meal has been previously reported and validated as a measure of disinhibited eating in youth [41,42].”

Line 159: Please add a manufacturer of the stadiometer.

Thank you for catching this oversight. On page 4, we have included the manufacturer information of the stadiometer: Holtain Ltd., Crymych, Wales, UK.

Please elaborate on the measurement of iDXA.

We have included the following sentence as a description of the iDXA procedure (page 4): “The iDXA scan is a non-invasive measurement that utilizes low-dose X-rays and requires the participant to lay flat on their back for 10-20 minutes as the scanning arm slowly moves across the body [45]”.

Discussion

Second paragraph, the first sentence: Please add a reference.

We apologize for this oversight. The appropriate citations have been added to the sentence on page 7, “The previous research on changes in food intake in adults during the pandemic have mostly been based on self-report surveys administered after the onset of the pandemic [13,15,29,30].”

A very important part in the prevention of obesity is physical activity. I think one of the reasons for the non-existence of differences between groups is the level of physical activity during the pandemic. Some participants may have practiced regular physical activity at home during the lockdown, which may influence their BMI, adiposity etc. You should mention that in the discussion. Also, I think this is one of the study’s limitations, you don’t have data on the physical activity level of the participants during the before and during the pandemic.

We agree with the reviewer that physical activity, or energy output, is a crucial piece of the energy balance equation and thus obesity prevention. We mention literature suggesting a reduction in physical activity in youth during the pandemic in the introduction, however we agree with the reviewer that this should have been noted in the Limitations section as well. In the third paragraph, second sentence of the Discussion section, we have made the following changes in order to recognize physical activity as an important factor and a limitation of the study (page 8):

“Our hypothesis that youth assessed during the pandemic would gain more weight and adiposity than youth examined before the pandemic was based on literature demonstrating that youth gaining excess weight shortly after the onset of the pandemic [10-17] was in part due to a reduction in their physical activity in the same time period [16,18-21].”

And, on page 8: “Our study did not assess changes in physical activity levels and it is possible that individual differences in energy output or sedentary activity during the pandemic could have affected the BMI and adiposity outcomes.”

Reviewer 3 Report

I had the opportunity to review this interesting study comparing the changes in BMI, body composition, BMIz, and energy intake between two groups of adolescents. The first group comprised individuals whose measurements were taken prior to the onset of the COVID-19 pandemic, while the second group consisted of individuals whose measurements were taken before and after the pandemic. The study utilizes a novel approach to evaluate these changes among adolescents during the COVID-19 pandemic, which is the study’s main strength. The study’s main limitation is that energy intake is evaluated in experimental conditions, which differs from the real-world situation. However, the authors have discussed this issue and how it may affect the findings well in the discussions. I have pointed out my comments below.

a.      Please add a brief description of the sampling procedure to the methods section.

b.      How did you determine the sample size? The sample size is relatively small, and recruiting 115 participants in your study needs justification. The lack of significant differences between groups might be due to the small sample size.

c.      According to Table 1, the mean age of participants was 12.4 years, which contrasts the text in which you mentioned the mean age of participants was 15.66 years. Please clarify. I think one refers to the baseline values and one to the age at the time of the second measurement.

d.      I suggest adding some information about the withing group changes in BMI, body composition, BMIz, and energy intake in both groups, which might help the readers understand the findings better.  

Author Response

Reviewer 3:

I had the opportunity to review this interesting study comparing the changes in BMI, body composition, BMIz, and energy intake between two groups of adolescents. The first group comprised individuals whose measurements were taken prior to the onset of the COVID-19 pandemic, while the second group consisted of individuals whose measurements were taken before and after the pandemic. The study utilizes a novel approach to evaluate these changes among adolescents during the COVID-19 pandemic, which is the study’s main strength. The study’s main limitation is that energy intake is evaluated in experimental conditions, which differs from the real-world situation. However, the authors have discussed this issue and how it may affect the findings well in the discussions. I have pointed out my comments below.

Thank you for your feedback and assessment of our manuscript.

Please add a brief description of the sampling procedure to the methods section.

Thank you for this suggestion. On page 3, the following information on recruitment efforts was added to the first paragraph of the Methods section: “Youth were generally health boys and girls ages 8-17 years at baseline recruited from the Washington D.C., Maryland, and Virginia via direct mailings to families within the greater metropolitan area of the Nationals Institutes of Health (NIH), flyers posted at the NIH and other local facilities (i.e., libraries, supermarkets, etc.), and advertisements and flyers posted online and on social media. All recruitment efforts were in the English language and targeted to families with youth in the appropriate age range.”

In the first sentence of the results section (page 5) we state, “Out of the 129 youth who completed a baseline and 3-year visit, 115 youth (89.2%) had complete data and were included in analyses (Table 1).” Additionally, we have added information on the excluded participants (page 5): “All 14 excluded participants were excluded for missing data; chi-squares found no significant differences in sex, race, ethnicity (ps=.15-.93) and t-tests found no significant differences in age, baseline BMI, and adiposity (ps=.18-.84) between excluded and included participants.”

We have included the following information in the third paragraph of the methods section to clarify the sampling procedure: “Sample selection was based on data completion and the dates of assessment. First, youth with incomplete data on any measures for either the baseline or 3-year visit were excluded. Second, youth were grouped into No Pandemic and Pandemic groups by the dates of their baseline and 3-year visits (Figure 1).”

At the time of the data analysis, all youth with complete data fit the requirements of the assessment dates, thus, all 115 youth were categorized into the No Pandemic and Pandemic groups.

How did you determine the sample size? The sample size is relatively small, and recruiting 115 participants in your study needs justification. The lack of significant differences between groups might be due to the small sample size.

The original study is ongoing in both the recruitment and measurement of participants. The study is expected to continue for at least 6 years after the recruitment of the final participant. The total enrollment goal is 500 healthy boys and girls. The sample size for this secondary analysis was based on the total number of participants with complete data on all measures used in the analyses at the time of the data analysis. The participants must also have been seen at least twice before the pandemic (No Pandemic group) or once before the pandemic and once during (Pandemic group), which limited the sample size. It is possible that the Pandemic group sample size will continue to grow as measurements continue, however the No Pandemic group sample size cannot be increased. Moreover, given that the impact of the pandemic has shifted considerably with vaccinations and the large number of people who have been exposed to the virus and thus most are no longer in lockdown or taking pandemic precautions, it is likely these factors might be confounders.

Nevertheless, we agree that the small sample size is a limitation of the study and we have added the following to the final paragraph of the discussion section (page 9): “Fourth, the sample size was limited to 115 youth due to an inability to collect additional data from No Pandemic children.”

According to Table 1, the mean age of participants was 12.4 years, which contrasts the text in which you mentioned the mean age of participants was 15.66 years. Please clarify. I think one refers to the baseline values and one to the age at the time of the second measurement.

Thank you for catching this error! We apologize and we have corrected the text to reflect the data presented in Table 1 (p. 5): “The sample had a mean age of 12.4 years at baseline (SD=2.7), was 47.8% female, 54.8% White, and 93% Non-Hispanic/Latino.”

I suggest adding some information about the within group changes in BMI, body composition, BMIz, and energy intake in both groups, which might help the readers understand the findings better. 

Thank you for this recommendation. We have now included a table with within group changes in the Supplemental Materials with a new subsection describing this information in the Results section on p. 6:

3.2 Within-Group Changes between Baseline and 3-Year Visits

        Difference scores were calculated for total energy intake, BMI, BMIz, and adiposity. Using paired-samples t-tests, the No Pandemic group had significant increases in energy intake and BMI (t(71)=-2.85, p=.005; t(71)=-6.96, p<.001) and the Pandemic group had significant increases in energy intake (t(42)=-3.35, p=.001), BMI (t(42)=-3.20, p=.003), and significant decreases in BMIz (t(42)=2.37, p=.021) and adiposity (t(42)=1.99, p=.050). See Supplemental Table S1. However, when the mean difference scores for each group were compared in a fully-adjusted GLM, there were no significant group differences in the within-group changes in total energy intake, BMI, BMIz, or adiposity. See Supplemental Tables S2-S3). These results reflect the natural and expected trend for youth to gain weight and consume more calories as they get older and gain height, and as the adjusted models demonstrate, when controlling for this growth trend, there were no significant differences in the change in total energy intake, BMI, BMIz, or adiposity between the baseline and 3-year visits in youth examined before the pandemic versus youth examined during the pandemic.

Supplemental Tables S1, S2, and S3 were added to the Supplemental Materials.

Round 2

Reviewer 3 Report

Thank you very much for revising the manuscript according to my comments. The manuscript's quality has improved after the revision, and I have no further comments.